# Estimating the optimal interval between rounds of indoor residual spraying of insecticide using malaria incidence data from cohort studies

Levicatus Mugenyi[1,2]*, Joaniter I. Nankabirwa[1,3], Emmanuel Arinaitwe[3], John Rek[3], Niel Hens[4,5], Moses Kamya[1,3], Grant Dorsey[6]

1 College of Health Sciences, Makerere University, Kampala, Uganda, 2 The Aids Support Organization, Kampala, Uganda, 3 Infectious Diseases Research Collaboration, Kampala, Uganda, 4 I-Biostat, Data Science Institute, Hasselt University, Hasselt, Belgium, 5 Centre for Health Economics Research and Modelling Infectious Diseases, Vaccine and Infectious Disease Institute, University of Antwerp, Antwerp, Belgium, 6 Department of Medicine, University of California, San Francisco, California, United States of America

* lmugenyi005@gmail.com

**Data Availability Statement:** Data for the study conducted from October 2011 through September 2017 (referred to as "PRISM1") can be found at

## Abstract

### Background

Indoor residual spraying (IRS) reduces vector densities and malaria transmission, however, the most effective spraying intervals for IRS have not been well established. We estimated the optimal timing interval for IRS using a statistical approach.

### Methods

Six rounds of IRS were implemented in Tororo District, a historically high malaria transmission setting in Uganda, during the study period (3 rounds with bendiocarb active ingredient (Ficam®): December 2014 to December 2015, and 3 rounds with pirimiphos methyl active ingredient (Actellic 300®CS): June 2016 to July 2018). A generalized additive model was used to estimate the optimal timing interval for IRS based on the predicted malaria incidence. The model was fitted to clinical incidence data from a cohort of children aged 0.5–10 years from selected households observed throughout the study period.

### Results

494 children, 67% aged less than 5 years at enrolment were analysed. Six-months period incidence of malaria decreased from 2.96 per person-years at the baseline to 1.74 following the first round of IRS and then to 0.02 after 6 rounds of IRS. The optimal time interval for IRS differed between bendiocarb and pirimiphos methyl and by IRS round. To retain an optimum impact, bendiocarb would require respraying 17 (95% CI: 14.2–21.0) weeks after application whereas pirimiphos methyl could remain impactful for 40 (95% CI: 37.0–42.8) weeks, although in the final year this estimates 36 (95% CI: 32.7–37.7) weeks. However, we could not estimate from the data the optimal time after the second and third rounds of bendiocarb and after the second round of pirimiphos methyl. Neither the amount of rainfall nor the EIR

https://clinepidb.org/ce/app/record/dataset/DS_0ad509829e. Data for the study conducted from October 2017 through October 2019 (referred to as "PRISM2") can be found at https://clinepidb.org/ce/app/record/dataset/DS_51b40fe2e2. A minimum data necessary to reproduce the findings is attached as Supporting Information file.

**Funding:** This research report is supported by the National Institute of Allergy and Infectious Diseases (NIAID) as part of the International Centers of Excellence in Malaria Research (ICEMR) program (U19AI089674) and the Fogarty International Center of the National Institutes of Health under Award Number D43TW010526. JIN is supported by the Fogarty International Center (Emerging Global Leader Award grant number K43TW010365.

**Competing interests:** The authors have declared that no competing interests exist.

nor the distribution of nets were found to be statistically significant for determining the time period between spray rounds.

## Conclusion

In our setting, the effect of the two IRS products was distinct. Statistically, pirimiphos methyl provided a longer window of protection than bendiocarb, although impact varied between different spray rounds and years which was not explained by rainfall or EIR or distribution of nets in our statistical approach. Understanding the effectiveness of IRS and how long it lasts can help for planning campaigns, but one should consider the financial cost and insecticide resistance. Monitoring the timing of spray campaigns using clinical incidence could be repeated in future programs to help determine the average period of protectivity of these products.

## Introduction

Despite recent efforts to scale-up coverage of malaria control interventions and the renewed focus on elimination, malaria remains a major global health problem [1]. However, marked declines in malaria burden have been documented in many sub-Saharan African counties linked to scale-up of vector control interventions including the use of insecticide treated bed-nets and indoor residual spraying (IRS) [1, 2]. Studies of malarial control in sub-Saharan African countries show that IRS, dramatically reduces malaria and its vectors [2, 3]. Similar to other African countries, IRS in Uganda has been shown to reduce the risk of parasitemia, anemia, malaria morbidity, vector densities and malaria transmission [4–7]. However, some IRS products have contrasting impacts [8]. For example, a study in Benin, West Africa shows that short lasting efficacy of Actellic 50 EC may render IRS ineffective [9].

The effectiveness of IRS may be affected by a number of factors including; the nature of the insecticide used, operational factors, malaria endemicity, and environmental factors [8]. The other factor which is often neglected is compliance by household owners to protect household space after IRS implementation to ensure high effectiveness [10]. One of the operational factors to consider is the timing interval for IRS when multiple rounds are sprayed. A study in Zimbabwe showed that bendiocarb remains active for up to 8 weeks with 96% mortality for mosquitoes on thatch [11]. The same study showed that bendiocarb remains active for up to 20 weeks with 74% mortality for mosquitoes on mud compared to 100% for mosquitoes on thatch [11]. In Madagascar, bendiocarb was shown to have a mortality inducing effect on local mosquitoes of up to 80% for up to 5 months post spraying [12]. Elsewhere in Zanzibar, a study to investigate the residual effect of pirimiphos methyl active ingredient sprayed on common surfaces of human dwellings showed that its mortality inducing effect on local mosquitoes was maintained on all sprayed surfaces up to 8 months post-IRS [13].

The World Health Organization (WHO) recommends scheduling IRS application to coincide with the build-up of vector populations just before the onset of the peak transmission season [14]. WHO further states that it is usually not operationally feasible to conduct more than two rounds of IRS in 1 year [14]. Although these guidelines highlight the importance of the timing of IRS, they do not provide the full information on the optimal interval for insecticides with different protective durations and assume a "one size fits all". In addition, the information on the effect of factors like seasonality and transmission intensity on the timing interval remain unclear.

In this paper, we provide estimates of optimal timing intervals for IRS with bendiocarb and pirimiphos methyl using malaria incidence data from a cohort of children aged 0.5–10 years living in a historically high malaria transmission area in Uganda. The use of incidence data allows us to test the durability of the spray campaigns on the clinical outcome. We apply a generalized additive model (GAM) to estimate the time-dependent malaria incidence while accounting for observed heterogeneity and later use this incidence to estimate the IRS timing interval. These time interval estimates could be useful to guide policy makers in similar settings on when the next round of IRS should be applied in order to sustain the effect of IRS towards elimination of malaria.

## Materials and methods

### Study setting

We used clinical data collected between October 2014 and March 2019 from a cohort of participants under the Program for Resistance, Immunology, Surveillance and Modelling of malaria (PRISM) project. The PRISM study was conducted in Nagongera sub-county Tororo district in Uganda, a traditionally high malaria transmission setting whose EIR was estimated at 310 infectious bites per person year in 2010 before the introduction of IRS [15]. Malaria transmission in Uganda is mostly perennial, so even though there are slight seasonal peaks, transmission continues all year making the durability of the IRS product an important consideration. IRS was introduced in Tororo for the first time in December 2014, and six rounds of IRS were applied during the study period. Three rounds of IRS with bendiocarb were applied approximately every six months (December 2014 –January 2015, June–July 2015 and November–December 2015), and three additional rounds of IRS with pirimiphos methyl were applied annually (June-July 2016, June-July 2017 and June-July 2018). The length of time in months it took to complete delivery of each spray campaign were 1.9 for round 1, 1.3 for round 2, 0.8 for round 3, 1.0 for round 4, 2.5 for round 5, and 1.3 for round 6. In May 2017, there was universal distribution of long-lasting insecticidal nets (LLINs) in Tororo as part of the National distribution campaign. Studies have shown an additional effect of LLINs over IRS [16]. The effect of LLINs on the IRS timing interval has been assessed in this analysis.

### Study population

The PRISM project had two phases. Phase I of the project has previously been described [15]. Briefly, the study ran between August 2011 and September 2017. In this phase, participants were recruited from 100 randomly selected households within the catchment area of Nagongera Health Centre IV. All children aged 6 months to 10 years and a primary adult caretaker in the selected household were enrolled. Participants were treated for all their health care needs and followed every 1–3 months to measure microscopic and sub-microscopic parasitemia at a dedicated clinic that was open 7 days a week. Clinical malaria was defined as having fever and a positive blood smear. Participants were treated at the study clinic for any febrile illness and symptomatic malaria was documented using passive surveillance. The cohort was dynamic and all children in the participating households that reached 6 months of age were enrolled and children greater than 10 years of age were excluded from further follow-up.

The second phase of PRISM was carried out between October 2017 and March 2019, four years after the role out of IRS. In this phase, 80 households including 33 households from PRISM phase 1 and 47 new randomly selected households. Unlike Phase 1, all household members of the selected households were enrolled into the cohort and followed at the dedicated clinic every four weeks. Follow-up procedure in the Phase 2 were similar to that of Phase 1. All participants in both phases of the study received a LLIN at enrolment. In order to work

with the same population from the two phases of PRISM, the analysis in this study was restricted to only children aged 0.5–10 years.

### Generalized additive model (GAM)

We used a GAM model to allow a non-parametric dependency of malaria incidence on time in weeks measured from each round of IRS and a parametric component to adjust for other covariates including age group, monthly rainfall (to adjust for seasonality), transmission intensity (using monthly entomological inoculate rate, EIR), and the duration (in months) since mosquito nets were distributed. EIR was calculated as a product of the daily human biting rate (HBR) and the sporozoite rate × 365 days/year as previously described by our group [17]. The GAM model was preferred over parametric models like generalised linear model (GLM) due to the non-linear and complex relationship between time and malaria incidence as shown in Fig 1. The outcome of interest was malaria incidence per person-years calculated for each week from the date of IRS application for each round. The incidence was obtained by dividing the number of malaria cases per week by total time at risk (person time) and was later converted into annual rates. The GAM model was formulated as follows [18].

Let $Y_{jk}$ be the outcome, which is malaria incidence per person-years at the $j^{th}$ week for the $k^{th}$ round of IRS; $t_{jk}$ be the $j^{th}$ week after the $k^{th}$ round of IRS modelled non-parametrically; $x_{jk}$ be a p × 1 vector of p-covariates for the parametric component. A GAM model then relates the

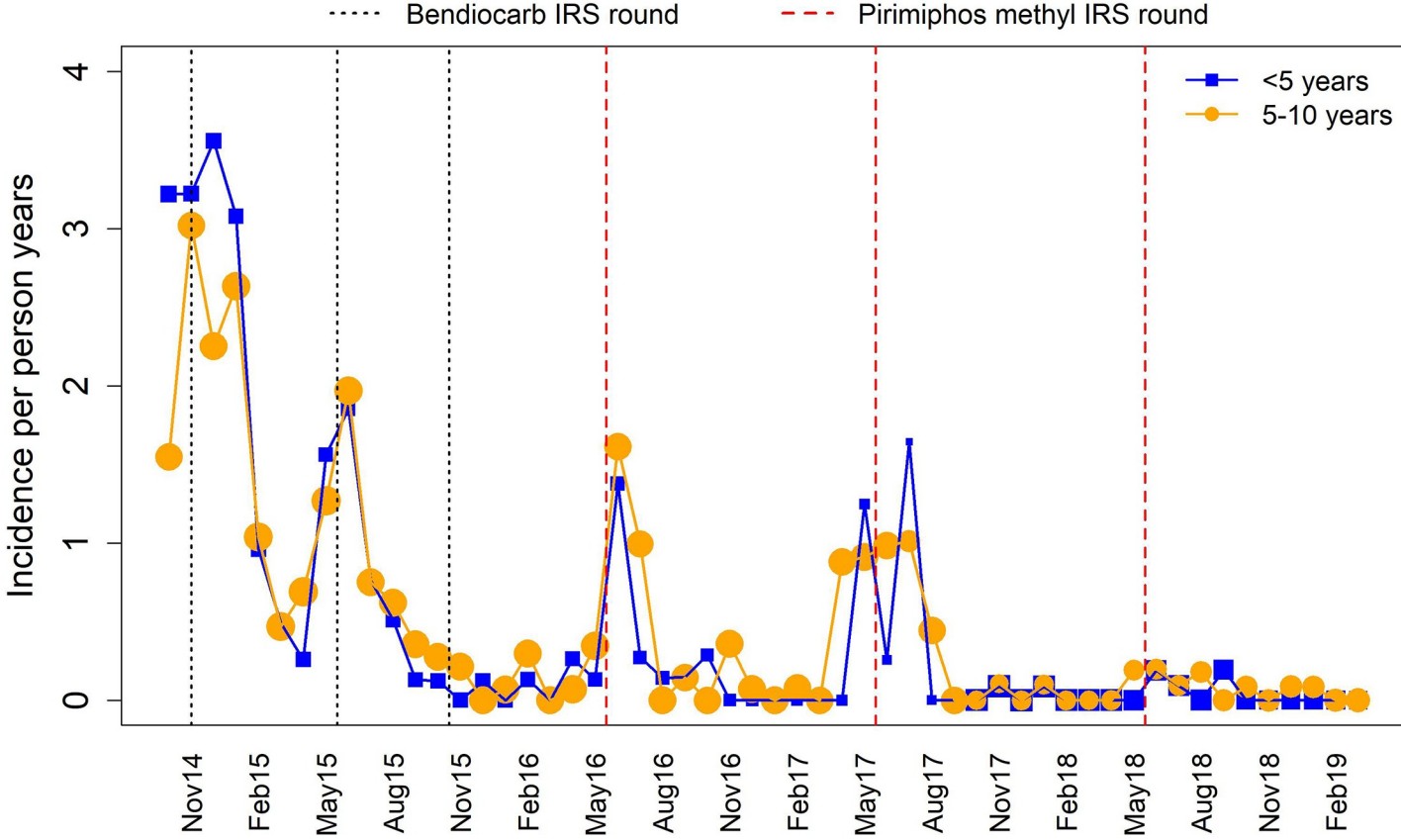

**Fig 1. Incidence of malaria per person-years among children between October 2014 (just before the first IRS round) and March 2019 following 6 rounds of IRS at a site of medium transmission in Uganda.** The vertical lines indicate time points at which the rounds of bendiocarb (dotted-black lines) and pirimiphos methyl (dashed-red lines) were applied, by age group. The size of the points corresponds to the number of children at risk.

mean $E(Y_{jk}) = \mu_{jk}$ to the time $t_{jk}$ and covariates $x_{jk}$, through a link function as follows [18];

$$g(\mu_{jk}) = x_{jk}{}^T\beta_k + \sum_{k=1}^{r} f_k(t_{jk}),$$  (1)

where $g(\mu_{jk})$ is a monotonic differentiable link function (e.g., identity in our case); $\beta_k$ is a vector of parameters associated with the parametric model components for the $k^{th}$ round of IRS; $f_k(t_{jk})$ is a centred twice differentiable smooth function for $k = 1,2,\ldots,r$ rounds of IRS.

Many techniques have so far been developed to estimate the smooth effects of the continuous variables in $f_k(t_{jk})$. In this paper, spline smoothing (also referred to as spline regression) is applied. For details about spline smoothing and GAM modelling, we refer the reader to Wahba [19], Reinsch [20], and Brumback & Rice [21].

For the IRS timing interval, we aim to obtain the value of time $t_{jk}$ for which we observe the first increase in the malaria incidence. Here forth, we refer to the time at which we observe the first increase in malaria incidence after each round of IRS as optimal time or optimal point.

During model building, we used a generalised cross-validation (GCV) score and deviance to check for model fit. A GAM with a smaller GCV score and higher explained deviance implied a better fit. For rainfall, we performed a sensitivity analysis considering 1-month, 2-months and 3-months lags.

## Ethics statement

All children aged 0.5–10 years who fulfilled the selection criteria and had written informed consent from a parent or guardian from each household were enrolled into the cohorts in Phase 1 and 2. Both phases of the PRISM study were approved by the Uganda National Council for Science and Technology (HS 1019 for Phase 1 and HS-119ES for Phase 2), Makerere University School of Medicine Research and Ethics Committee (2011–167 for Phase 1 and 2017–099 for Phase 2), the University of California, San Francisco Committee on Human Research (11–05995 for Phase 1 and 17–22544 for Phase 2).

## Results

In total, 494 children aged 0.5–10 years contributed to the clinical data observed between October 2014 and March 2019, with a majority (67.0%) less than 5 years of age at enrolment. Table 1 shows number of children at risk and malaria incidence per person-year for 10 six-months periods (2 baseline and 8 post IRS) and 1 five-months period (after the last round of IRS), presented overall and by age group. Malaria incidence decreased from 3.07 per person-year during the first baseline period (November 2013-April 2014) to 2.96 during the second baseline period (May-October 2014) and then to 1.74 during the first period following the first round of IRS application (November 2014-April 2015). The incidence declined to 0.02 per person-year during the last period following the $6^{th}$ round of IRS. The incidence was higher among children aged less than 5 years during the two baseline periods and the period following the first round of IRS, after which it was consistently higher among those aged 5–10 years.

Fig 1 shows monthly incidence of malaria for the period running from October 2014 to March 2019 indicating different points of application of the two types of IRS insecticides (bendiocarb and pirimiphos methyl). We observe down and upward trends in the incidence after each round of IRS with no clear difference between children aged less than 5 and those aged 5–10 years. We also observe transmission peaks in the months of May and June which coincided with the second, fourth, fifth and the sixth rounds of IRS.

Though the 2-month lag of rainfall fitted the data better than 1-month and 3-month lags, it did not improve the overall fit (p = 0.105) and it was dropped. The other variables that did not improve model fit and were dropped from the model included age group (p = 0.270), monthly

**Table 1. Number at risk and six-months period malaria incidence comparing baseline and post IRS periods.**

|  | Overall | <5 years[¶] | 5–10 years[¶] |
|---|---|---|---|
| **November 2013 –April 2014[b]** | **n = 290** | **n = 125** | **n = 180** |
| Person years | 137.59 | 54.64 | 82.95 |
| Incidence per person-year | 3.07 | 3.95 | 2.50 |
| **May 2014 –October 2014 [b]** | **n = 289** | **n = 118** | **n = 189** |
| Person years | 139.95 | 54.94 | 85.01 |
| Incidence per person-year | 2.96 | 3.40 | 2.67 |
| **November 2014 –April 2015** | **n = 282** | **n = 108** | **n = 190** |
| Person years | 136.81 | 48.0 | 88.81 |
| Incidence per person-year | 1.74 | 1.94 | 1.63 |
| **May 2015 –October 2015** | **n = 288** | **n = 104** | **n = 189** |
| Person years | 136.86 | 47.39 | 89.48 |
| Incidence per person-year | 0.83 | 0.80 | 0.85 |
| **November 2015 –April 2016** | **n = 279** | **n = 103** | **n = 188** |
| Person years | 136.35 | 48.17 | 88.18 |
| Incidence per person-year | 0.095 | 0.08 | 0.10 |
| **May 2016 –October 2016** | **n = 275** | **n = 94** | **n = 194** |
| Person years | 130.70 | 43.05 | 87.66 |
| Incidence per person-year | 0.46 | 0.39 | 0.49 |
| **November 2016 –April 2017** | **n = 210** | **n = 68** | **n = 152** |
| Person years | 4.17 | 1.35 | 2.82 |
| Incidence per person-year | 4.56 | 0.00 | 6.73 |
| **May 2017 –October 2018** | **n = 365** | **n = 170** | **n = 203** |
| Person years | 12.61 | 5.49 | 7.12 |
| Incidence per person-year | 5.87 | 3.10 | 8.00 |
| **November 2018 –April 2018** | **n = 266** | **n = 150** | **n = 131** |
| Person years | 126.2 | 67.19 | 59.01 |
| Incidence per person-year | 0.03 | 0.03 | 0.03 |
| **May 2018 –October 2018** | **n = 267** | **n = 139** | **n = 145** |
| Person years | 129.41 | 64.15 | 65.26 |
| Incidence per person-year | 0.10 | 0.08 | 0.12 |
| **November 2018 –March 2019** | **n = 271** | **n = 131** | **n = 152** |
| Person years | 129.41 | 48.82 | 57.58 |
| Incidence per person-year | 0.02 | 0.00 | 0.03 |

[b] Baseline period

[¶] *cohort age used for the age groups, n = number at risk.*

EIR (p = 0.218) and the duration in months since nets were distributed (p = 0.249). The final fitted GAM included only the intercept for the parametric component and the splines for each round of IRS for the non-parametric component with explained deviance of 80.8% and a minimized GCV score of 0.133.

Table 2 shows the parameter estimates, standard errors and corresponding test results of the fitted GAM. The estimated effective degrees of freedom for the smooth terms differed by IRS round and were each different from 1, implying that a linear relationship was not feasible in our case. A diagnostic plot showing residuals against time with the estimated smooth parameters is given in Fig 2. The horizontal lines around 0 indicate better fits for the smoothers for all rounds of IRS except for round 1 and 2 of bendiocarb, which is due to a smaller number of children that were at risk in the last weeks.

**Table 2. Estimates of the fitted GAM model using B-splines.**

| Effect | | Estimate (SE) | t-value | p |
|---|---|---|---|---|
| Intercept | | 0.30 (0.03) | 9.42 | <0.001 |
| **Smooth effects** | | **Effective degrees of freedom** | **F-value** | **p** |
| Time since IRS round (weeks): | | | | |
| Bendiocarb | Round 1 | 7.00 | 77.75 | <0.001 |
| | Round 2 | 4.59 | 16.17 | <0.001 |
| | Round 3 | 1.79 | 3.88 | 0.019 |
| Pirimiphos methyl | Round 4 | 6.99 | 12.16 | <0.001 |
| | Round 5 | 4.41 | 14.99 | <0.001 |
| | Round 6 | 4.82 | 2.35 | 0.037 |

Fig 3 shows observed and predicted weekly malaria incidence for each round of the two types of insecticides (bendiocarb and pirimiphos methyl). The figure also shows the amount of rainfall in mm recorded at different calendar months (vertical bars). The vertical dotted lines in Fig 3 indicate time points at which we first observe an increase in malaria incidence following a decreasing trend after each round of IRS. These points refer to optimal time for applying another round of IRS. Results suggest that the optimal time for applying another round of bendiocarb after the first round was 17 weeks (95% confidence interval (CI): 14.2–21.0) (about 4 months) and 40 weeks (95% CI: 37.0–42.8) (about 10 months) after the first round of pirimiphos methyl. The optimal time for applying another round of pirimiphos methyl after the third round (round 6 overall) was 36 weeks (95% CI: 32.7–37.7) (about 9 months). After the second rounds of bendiocarb and pirimiphos methyl (round 5 overall), the incidence

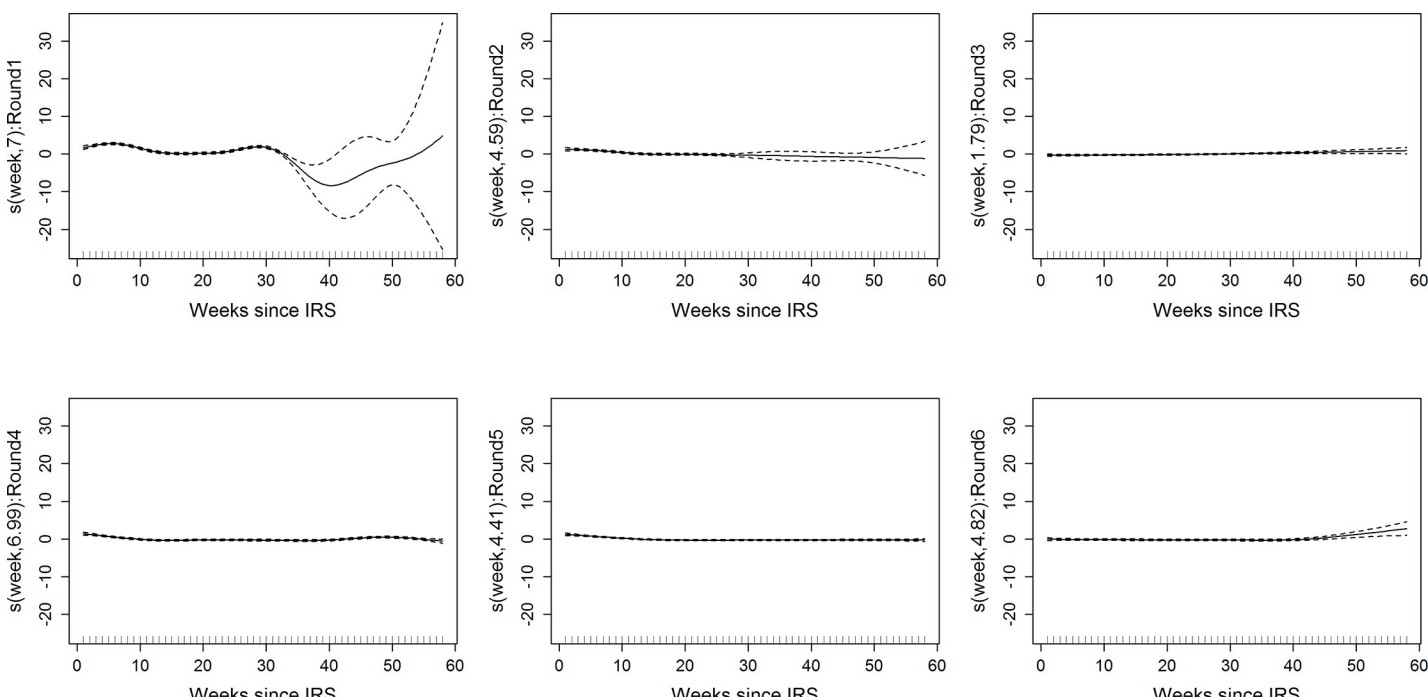

**Fig 2.** Diagnostic plots showing residuals for the fitted B-splines with 95% credible regions (dotted lines) verses time (in weeks) after each round of bendiocarb (top row) and pirimiphos methyl (bottom row) insecticides using different degrees of freedom (in brackets on the y-axis). A straight horizontal line through point 0 on the y-axis implies better fit.

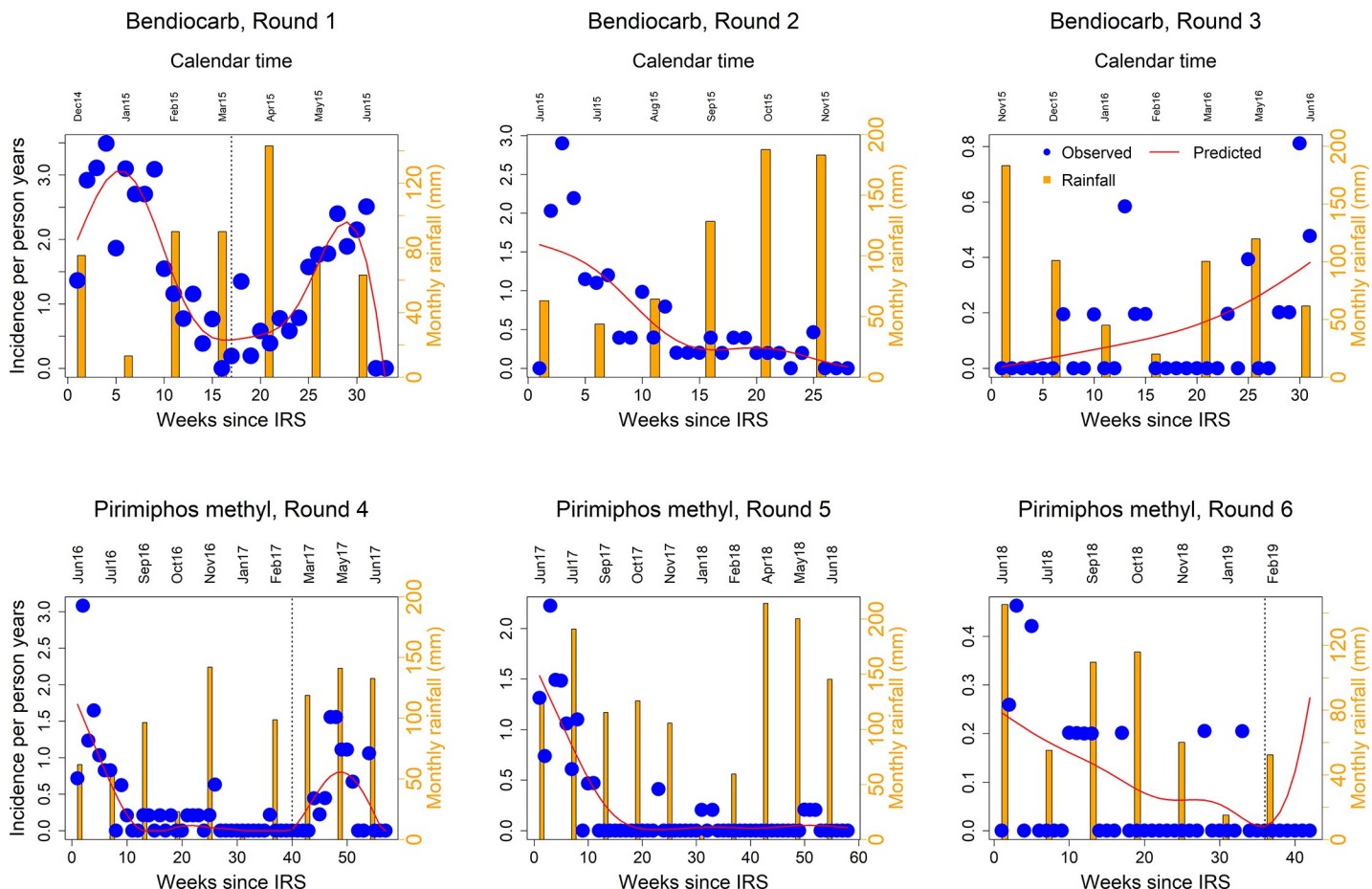

**Fig 3.** Observed (blue points) and predicted (solid red line) incidence of malaria per person-years against time in weeks since application of IRS and monthly rainfall (vertical orange bars) against calendar time for each round of bendiocarb (top row) and pirimiphos methyl (bottom row). The secondary y-axis labelled in orange represents the amount of monthly rainfall in mm. The vertical dotted lines indicate optimal time for applying another round of IRS.

continuously decreased and the minimum point could not be established. After the third round of bendiocarb, unlike the other rounds of IRS, the incidence continuously increased and we were not able to estimate the optimal time. The figure also indicates that malaria incidence after each round of IRS does not depend on the amount of rainfall and consequently the optimal time for IRS.

## Discussion

Our study suggests that for effective malaria control, the optimal time for applying another round of IRS after the first one is 17 weeks for bendiocarb and 40 weeks for pirimiphos methyl. After the third round of pirimiphos methyl, a slightly shorter interval of 36 weeks was found optimal before another round. We could not establish from the data the optimal time for applying another round of IRS after the second and the third rounds of bendiocarb as well as after the second round of pirimiphos methyl due to failure to attain the minimum points. We used a generalized additive model (GAM) to estimate malaria incidence, where time in weeks from each round of IRS was modelled non-parametrically and later used the time-dependent incidence to obtain optimal time for IRS application. We applied the model to data from a

cohort study of children aged 0.5–10 years from an area of previously high transmission (Tororo district) in Uganda [15].

Though our findings agree that pirimiphos methyl is longer-acting than bendiocarb [22, 23], the results suggest shorter intervals than those claimed by the manufactures of these insecticides [23] and as applied during the study implementation [24]. For example, instead of considering 6 months (24 weeks) after each round of bendiocarb as was applied during the study, a shorter interval of about 4 months after the first round could be appropriate. Also, instead of the 12 months interval after each round of pirimiphos methyl which was the case during the study, the findings suggest about 10 months for the second and 9 months for the fourth round. These findings also suggest that for bendiocarb, more than 2 rounds of IRS in a year could be required which contradicts with WHO recommendation of not more than two rounds per year [14]. However, for pirimiphos methyl, the results suggest that about one round per year is feasible and this concurs with the WHO recommendations [14]. The timing intervals suggested by our results also concurs with results by the study in the Northern part of Uganda (period 2012–2015) where incidence in the seventh month following IRS was almost twice that of the last spray month with IRR of 1.75 [25]. However, the authors did not specify the type of insecticide used. This implies that waiting for 7 months in case of bendiocarb before another round of IRS may leave communities unprotected.

The finding on increase in malaria incidence after IRS application is similar to what other authors reported. For example, other studies in Uganda documented increases in malaria incidence [25] and test positivity rate (TPR) post IRS [24, 26]. This could be due to poor timing of IRS relative to the transmission peak. For example, our results indicate higher transmission peak in May/June time, so perhaps the November spray rounds may have artificially bigger impacts due to the natural decline in transmission observed during these months of the year.

The findings show that the timing interval was not altered by the amount of rainfall, the entomological inoculation rate (EIR), and the duration in months from when nets were distributed. For rainfall and EIR, perhaps it is because the data we used were already aggregated by month. Finer intervals like weekly data would help to improve our estimates but were not available.

This work highlights well the variability in the length of time that the active ingredient deployed during a spray campaign may have, even for the same product within a single setting. Real-time surveillance of trends in clinical incidence could be used to help advise strategy teams on when to re-deploy the IRS intervention, ensuring rotation of products as guided by insecticide resistance management policies [14].

The findings in this study have four limitations. First, the results presented here were limited to one site, hence conclusions may not be generalized. Secondly, the study was not designed to estimate the timing interval for IRS. A future study designed to estimate the timing interval for IRS could help improve our estimates. Thirdly, our analysis did not consider asymptomatic infections which could possibly alter our estimates. For example, in the study setting the prevalence of asymptomatic infection was estimated at 58% in October 2014 (baseline) and 7% in February 2019 (end line) [17]. Fourth, we assume no change in the level of insecticide resistance to either product throughout the time-series. We did not have susceptibility bioassays for the study region to corroborate this assumption.

In conclusion, for effective control of malaria using IRS, different spraying intervals are needed for different rounds of bendiocarb and pirimiphos methyl insecticides. For example, the second round of bendiocarb needs to be applied 17 weeks (about 4 months) from the first round, and for pirimiphos methyl, 40 weeks (10 months) for round two, and 36 weeks (9 months) for round four. These intervals were not altered by the amount of rainfall, EIR, and the duration in months since nets were distributed. Though shorter intervals for IRS than the

usual practice are suggested by this study, cost-benefit analysis should be practiced to balance between effectiveness of IRS and implementation costs and insecticide resistance.

## Supporting information

**S1 File.**
(CSV)

**S2 File.**
(CSV)

## Author Contributions

**Conceptualization:** Levicatus Mugenyi.

**Data curation:** Levicatus Mugenyi.

**Formal analysis:** Levicatus Mugenyi, Joaniter I. Nankabirwa, Grant Dorsey.

**Funding acquisition:** Moses Kamya.

**Investigation:** Joaniter I. Nankabirwa, Emmanuel Arinaitwe, John Rek, Moses Kamya, Grant Dorsey.

**Methodology:** Levicatus Mugenyi, John Rek, Niel Hens, Grant Dorsey.

**Resources:** Moses Kamya.

**Software:** Levicatus Mugenyi.

**Supervision:** Niel Hens, Moses Kamya, Grant Dorsey.

**Validation:** Levicatus Mugenyi, Joaniter I. Nankabirwa, Emmanuel Arinaitwe, John Rek, Niel Hens, Moses Kamya, Grant Dorsey.

**Visualization:** Levicatus Mugenyi, Grant Dorsey.

**Writing – original draft:** Levicatus Mugenyi.

**Writing – review & editing:** Levicatus Mugenyi, Joaniter I. Nankabirwa, Emmanuel Arinaitwe, John Rek, Niel Hens, Moses Kamya, Grant Dorsey.

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
