## [Decision Letter · Decision Letter 0]

7 Apr 2020

PONE-D-20-03773

Estimating the Timing Interval for Malarial Indoor Residual Spraying: A Modelling Approach

PLOS ONE

Dear Dr Mugenyi,

Thank you for submitting your manuscript to PLOS ONE. After careful consideration, we feel that it has merit but does not fully meet PLOS ONE’s publication criteria as it currently stands. Therefore, we invite you to submit a revised version of the manuscript that addresses the points raised during the review process.

We would appreciate receiving your revised manuscript by May 22 2020 11:59PM. To enhance the reproducibility of your results, we recommend that if applicable you deposit your laboratory protocols in protocols.io, where a protocol can be assigned its own identifier (DOI) such that it can be cited independently in the future. For instructions see: http://journals.plos.org/plosone/s/submission-guidelines#loc-laboratory-protocols

We look forward to receiving your revised manuscript.

Kind regards,

David A. Larsen, PhD

Academic Editor

PLOS ONE

Journal Requirements:

2. Please provide additional details regarding participant consent. In the ethics statement in the Methods and online submission information, please ensure that you have specified (1) whether consent was suitably informed and (2) what type you obtained (for instance, written or verbal). Since your study included minors under age 18, state whether you obtained consent from parents or guardians. If the need for consent was waived by the ethics committee, please include this information.

"This research report is supported by the National Institute of Allergy and Infectious Diseases (NIAID) as part of the International Centers of Excellence in Malaria Research (ICEMR) program (U19AI089674) and the Fogarty International Center of the National Institutes of Health under Award Number D43TW010526. JIN is supported by the Fogarty International Center (Emerging Global Leader Award grant number K43TW010365). The content is solely the responsibility of the authors and does not necessarily represent the official views of the National Institutes of Health."

 "LM Received funding from Fogarty International Center of the National Institutes of Health under Award Number D43TW010526. URL of the funder https://www.fic.nih.gov/Pages/Default.aspx. The funder did not play any role in the the study design, data collection and analysis, decision to publish, or preparation of the manuscript"

5. Your ethics statement must appear in the Methods section of your manuscript. If your ethics statement is written in any section besides the Methods, please move it to the Methods section and delete it from any other section. Please also ensure that your ethics statement is included in your manuscript, as the ethics section of your online submission will not be published alongside your manuscript.

Additional Editor Comments (if provided):

Please in your revision can you clarify the methods somewhat. The STROBE guidelines and outlines can be useful.

Reviewers' comments:

Reviewer's Responses to Questions

**Comments to the Author**

1. Is the manuscript technically sound, and do the data support the conclusions?

Reviewer #1: Partly

Reviewer #2: Partly

Reviewer #3: Partly

2. Has the statistical analysis been performed appropriately and rigorously? 

Reviewer #1: Yes

Reviewer #2: Yes

Reviewer #3: No

3. Have the authors made all data underlying the findings in their manuscript fully available?

Reviewer #1: Yes

Reviewer #2: Yes

Reviewer #3: Yes

4. Is the manuscript presented in an intelligible fashion and written in standard English?

Reviewer #1: Yes

Reviewer #2: Yes

Reviewer #3: Yes

5. Review Comments to the Author

Reviewer #1: General comments

This article is timely and addresses a very important neglected component of IRS that can effectively improve its impact, and it is definitely worth publishing and communicating the information to the implementing programs. Even though the author has addressed most of the relevant points, the following aspects, comments and questions need clarification before considered for publishing this article.

Introduction:

• The authors have explained that IRS has been shown to be effective in terms of reducing malaria incidences and entomological indicators in various settings. However, in several other studies conducted from other endemic settings across Africa, contrasting results have also been documented, and this does not devalue the importance of IRS. So I suggest to the authors to include the two contrasts in their paper and perhaps, as timing could partly play a role in the negative outcomes reported in other endemic settings alongside other factors.

• The authors have indicated factors that in general may affect the overall effectiveness of IRS such as individual insecticides, operational factors etc., however, one of the key indicator or factor crucial to the effectiveness of IRS is compliance even though extremely neglected for IRS, which is basically protecting the personal household space after IRS implementation to ensure high effectiveness of the product. I also suggest here to the authors to include human behavioural factor as various activities conducted by household owners within personal household spaces may greatly affect the IRS efficacy and hence needs to be considered as a key component while planning for IRS implementation periods.

Methodology:

• LLINs were distributed almost less than 2 months before IRS implementation, in 2017, and given that nets (pyrethroid nets) have been shown to repel or irritate mosquitoes, is there any masking effect that may have led to the assumption that incidence greatly reduced as a result of IRS?

Reviewer #2: This is an interesting analysis of clinical incidence data tracking a cohort of under 10-year olds living in a high transmission setting that has implemented IRS. During Phase I, the study explores the use of bendiocarb sprayed approximately bi-annually. During phase II, the study explores annual implementation with Actellic 300®CS. The analysis is robust, and this is an interesting method to determine the window between spray rounds. However, I have some major concerns for the authors to consider prior to acceptance for publication.

Major concerns

1. The authors do not mention the natural seasonality of mosquitoes, how their density may change in the absence of chemical intervention, or whether these changes subsequently drive any seasonality in clinical incidence. Please could the authors comment on how this was accounted for in the statistical model as an increase in cases (the optimal point) could still indicate a working IRS product if the cases increase at a slower rate than expected in the absence of said product at specific times of the year. Were there any data on mosquito vector densities to corroborate the impact on incidence is driven by the IRS impact?

2. How was clinical incidence monitored in the trial? Were rapid diagnostic tests used to confirm positivity? This is important because if the presence of fevers are used to indicate malaria positivity (the authors note symptoms, but do not specify what these are for this study) then ‘true’ seasonality may be dampened because of other infections causing fever that are mis-diagnosed to be malaria. Please comment to clarify.

3. As all participants also received an ITN on enrolment, how were the impacts of these interventions decoupled from the impact of the IRS products? How were the effects adjusted for later rounds of IRS when the ITN efficacy would have waned and adherence to ITN use may have diminished?

In addition, were ITNs redistributed at any point through the study? If not, is the impact from Actellic even greater because it acts with older ITNs – potentially used less by the community and almost certainly working less effectively because the active ingredient has waned. It is critical to comment or account for this effect.

4. Given that malaria cases are so sensitive to the local ecology for any given transmission season, could the authors comment on the uncertainty in their suggested interval for spray rounds and whether they can infer anything from the statistical model about how this window may vary season to season, year to year or across any other boundary. Is it possible to make any sort of prediction for other settings using this framework? The authors note that uncertainty will be considered in future work but this should be included here. Could you infer uncertainty from the 90% credible intervals of the Bayesian simulations carried through from the splines? Or use the mean +/- 1.96*SE to estimate some uncertainty for the spline predictions shown in Fig 3?

Is there any sense in fitting the estimate across rounds for each chemistry? Rather than after each round separately to try and infer a more generalised estimate for this interval between spraying? Could the authors source data from other sites to refit the model with additional data which would enable the uncertainty to be better captured and results to be generalised which would be more useful for guiding policy decisions.

Please see additional comments in the attached document.

Reviewer #3: Manuscript: Estimating the Timing Interval for Malarial Indoor Residual Spraying: A Modelling Approach

The authors provided some interesting data for effective use IRS in malaria control, thought there are issues that need attention

One is the cost effectiveness of the repeated application of the IRS, which is also directly related with the sustainability of the program

Another one is that the authors indicated the malaria incidence at baseline month, midline month and endline month, but not in between. Fig 1 shows some interesting results—where more malaria cases are observed immediately after IRS of both insecticides.

What would happen if the first application miss correct timing—which month/or season is suitable for the first IRS? The first IRS timing not affect the results?

What would happen if the actellic insecticide is the only insecticide applied throughout the study period? --- 40 weeks intervals still works? What about the bediocarb alone?

The authors didn’t indicate the impact of bed nets independently? Bed nets could affect the interval?

Fig 2 and 3---hard to understand— possible to simplify

Methods: Was it systematically designed to answer the question? For example – the number of children in phase 1 and 2 are different? This is weakest part of this study, and lead to the results of inconclusive.

Discussion: ….The finding on increase in malaria incidence after IRS application is similar to what other authors reported. For example, other studies in Uganda documented increases in malaria incidence and test positivity rate (TPR) post IRS…. not justifiable results

This result need another justification ….method section and the initial spray time

Conclusions: there are a lot of uncertainty to reach these conclusions

Finally, major revision is required if considered for publication

6. PLOS authors have the option to publish the peer review history of their article (what does this mean?). If published, this will include your full peer review and any attached files.

Reviewer #1: No

Reviewer #2: No

Reviewer #3: Yes: Fekadu Massebo (Dr)

---

## [Author Response · Author response to Decision Letter 0]

15 May 2020

Response: Attention has been paid to ensure the manuscript meets style requirements 

2. Please provide additional details regarding participant consent. In the ethics statement in the Methods and online submission information, please ensure that you have specified (1) whether consent was suitably informed and (2) what type you obtained (for instance, written or verbal). Since your study included minors under age 18, state whether you obtained consent from parents or guardians. If the need for consent was waived by the ethics committee, please include this information.

Response: The consenting statement has been added and the ethical section has been placed under methodology. The following section on ethics has been included in the manuscript and web submission. Lines 157-164 of manuscript.

“Ethics statement

All children aged 0.5–10 years who fulfilled the selection criteria and had written informed consent from a parent or guardian from each household were enrolled into the cohorts in Phase 1 and 2. Both phases of the PRISM study were approved by the Uganda National Council for Science and Technology (HS 1019 for Phase 1 and HS-119ES for Phase 2), Makerere University School of Medicine Research and Ethics Committee (2011-167 for Phase 1 and 2017-099 for Phase 2), the University of California, San Francisco Committee on Human Research (11-05995 for Phase 1 and 17-22544 for Phase 2). ”

Response: The minimal anonymized data set necessary to replicate our study findings has been added as supporting information file.

"This research report is supported by the National Institute of Allergy and Infectious Diseases (NIAID) as part of the International Centers of Excellence in Malaria Research (ICEMR) program (U19AI089674) and the Fogarty International Center of the National Institutes of Health under Award Number D43TW010526. JIN is supported by the Fogarty International Center (Emerging Global Leader Award grant number K43TW010365). The content is solely the responsibility of the authors and does not necessarily represent the official views of the National Institutes of Health."

 "LM Received funding from Fogarty International Center of the National Institutes of Health under Award Number D43TW010526. URL of the funder https://www.fic.nih.gov/Pages/Default.aspx. The funder did not play any role in the study design, data collection and analysis, decision to publish, or preparation of the manuscript"

Response: The funding-related text is now removed from the manuscript. Please revise the funding statement to read as follows: “This research report is supported by the National Institute of Allergy and Infectious Diseases (NIAID) as part of the International Centers of Excellence in Malaria Research (ICEMR) program (U19AI089674) and the Fogarty International Center of the National Institutes of Health under Award Number D43TW010526. JIN is supported by the Fogarty International Center (Emerging Global Leader Award grant number K43TW010365)”

5. Your ethics statement must appear in the Methods section of your manuscript. If your ethics statement is written in any section besides the Methods, please move it to the Methods section and delete it from any other section. Please also ensure that your ethics statement is included in your manuscript, as the ethics section of your online submission will not be published alongside your manuscript.

Response: Done. The ethics statement has been transferred under methods section and deleted where it was placed just before reference. Lines 157-164 of manuscript.

---

## [Decision Letter · Decision Letter 1]

1 Jul 2020

PONE-D-20-03773R1

Estimating the Optimal Interval between Rounds of Indoor Residual Spraying of Insecticide Using Malaria Incidence Data from Cohort Studies

PLOS ONE

Dear Dr. Mugenyi,

Thank you for submitting your manuscript to PLOS ONE. After careful consideration, we feel that it has merit but does not fully meet PLOS ONE’s publication criteria as it currently stands. Therefore, we invite you to submit a revised version of the manuscript that addresses the points raised during the review process.

We look forward to receiving your revised manuscript.

Kind regards,

David A. Larsen, PhD

Academic Editor

PLOS ONE

Additional Editor Comments (if provided):

Dear Author, please see the following comments from Reviewer 2. They submitted these in a word document.

Review

Article: PONE-D-20-03773R1

Title: Estimating the Optimal Interval between Rounds of Indoor Residual Spraying of Insecticide Using Malaria Incidence Data from Cohort Studies

Summary: The work tracks clinical incidence data for a cohort of 6-month to 10-year olds living in a high transmission setting that has implemented IRS. During Phase I, the study explores the use of bendiocarb sprayed approximately bi-annually. During phase II, the study explores annual implementation with Actellic 300®CS. The limitations of the study are better explained in this revision. However, I still have some major reservations.

There remain a few queries on the methods and I have some hesitations on how the results are discussed – particularly in the abstract

Thank you for providing the data, that is useful for the review process. How was the annual EIR calculated? Please add a few sentences to explain how this was done with the data available. Similarly, it looks like rainfall was estimated monthly or over longer periods – is it possible to get finer scale information on rainfall – by week perhaps? This could impact dramatically on whether it is statistically important.

The methods first focus on the statistical methods, but it may be more intuitive to present the study area, data collection and PRISM trial prior to the statistics to help the reader follow the work reported.

Have you tried to include a variable like ‘month since net distribution’ or even a binary ‘net distribution within 1 year’ variable to try to account for the distinct impact seen from new nets?

The authors present, in the abstract, the clinical incidence from October 2014 and February 2019. But this could be driven by seasonality. It would be better to compare two times from the same month of the year, or even better still, averaging clinical incidence across a wider window of time to mitigate somewhat for natural between-year stochasticity in the metric. For how many months were the children tracked prior to application of the spray? Could you average across this entire period and compare it to the same length of time (and months) in the post application periods? Can you also provide the baseline data in the supplement?

I have the same issue with Table 1.

Table 1 is not so useful as it currently stands – it could be more informative to average or sum clinical incidence in the tracked cohorts across longer time windows (matching months) as, although seasonal rainfall did not appear to be associated with the duration of IRS effectivity in the GAM analysis, seasonal patterns in malaria incidence are likely and do tend to drive transmission by increasing the EIR, so comparing different times of the year may be misleading. Again, averaging or incidence over 3 or 6 months at baseline and comparing that to the same months in later years would be a fairer presentation. This seems particularly important because Fig 1 seems to suggest a higher mean incidence in Oct 2016 (than the baseline Oct 2014 estimate) but low in Dec 2016 – which is chosen as the comparison estimate … this is not a fair comparison so estimating the average clinical incidence across multiple months to some extent could overcome this potential stochasticity in measurements and alleviate the need to arbitrarily select a single measurement for the table.

I have also a suggestion for the abstract as I think that the methods/results and particularly conclusions paragraphs could be more useful if altered slightly.

• Use the IRS product (presumably Ficam® and Actellic 300®CS) or the insecticide active ingredient (bendiocarb and pirimiphos methyl) not one of each [this issue applies throughout the manuscript]

• Present the empirical data first, then the statistics

• Refer to the model as a statistical approach (as it is not a predictive model as the mechanism of transmission is not determined), so statistical approach is clearer.

• My greatest hesitation is with the conclusion. Here, a statistical model is applied which is useful to capture the direction of an effect or compare the two IRS products, but it is not useful for future projections because it does not capture the mechanism of impact. So, inferring from the statistical model needs to be done with care. Drawing contrasts between the spray applications is reasonable, but perhaps soften how this is stated so as not to determine future application.

Below is a suggestion using tracked changes [comments to address/delete!]:

“Background : Indoor residual spraying (IRS) reduces vector densities and malaria transmission, however, the most effective spraying intervals for IRS have not been well established. We estimated the optimal timing interval for IRS using a statistical approach.

Methods : Six rounds of IRS were implemented in Tororo District, a historically high malaria transmission setting in Uganda, during the study period (3 rounds with Ficam (bendiocarb active ingredient): December 2014 to December 2015, and 3 rounds with Actellic 300®CS (pirimiphos methyl active ingredient): June 2016 to July 2018). Generalized additive models were used to estimate the optimal timing interval for IRS based on the predicted malaria incidence. The model was fitted to clinical incidence data from a cohort of children aged 0.5–10 years from selected households observed throughout the study period. Annual entomological inoculation rates were estimated [how] and aggregated monthly [looking at the data file it looks like these were averaged?] .

Results : Monthly incidence of malaria from October 2014 to February 2019 decreased from 3.25 to 0.0 per person-years in the children under 5 years, and 1.57 to 0.0 for 5-10 year-olds [this is not appropriate, please see comments above]. The optimal time interval for IRS differed between bendiocarb and pirimiphos methyl and by IRS round. To retain an optimum impact, Ficam® would require respraying 17 weeks after application whereas Actellic 300®CS could remain impactful for 40 weeks, although in the final year this estimate 36 weeks. However, we could not estimate from the data the optimal time after the second and third rounds of bendiocarb and after the second round of actellic. Neither the amount of rainfall nor the EIR were found to be statistically significant for determining the time period between spray rounds [although these are aggregated by month(?) so the data may lack the scale needed to identify impact].

Conclusion: In our setting, the effect of the two IRS products was distinct. Statistically, Actellic 300®CS provided a longer window of protection than Ficam®, although impact varied between different spray rounds and years which was not explained by rainfall or EIR in our statistical approach. Understanding the effectiveness of IRS and how long it lasts can help for planning campaigns, but one should consider the financial cost and insecticide resistance. Monitoring the timing of spray campaigns using clinical incidence could be repeated in future programs to help determine the average period of protectivity of these products.”

There is no discussion on the timing of the intervention relative to the transmission peak. Generally, Uganda has a higher transmission peak in May/June time, so perhaps the November spray rounds have artificially bigger impacts due to fewer cases / less transmission in these periods of the year.

Thank you for adding the sentence on fever and blood smear positivity. Just to clarify, did the blood smear have to be positive for a clinical case to be scored? Or were some patients classed to be malaria positive by fever only? This could alter assumptions.

It may be worth including a sentence or two on what it means for the analysis to miss asymptomatic cases and what proportion of the population may be asymptomatic (a good paper on this issue e.g. Lindsay Wu et al Nature 2015: https://www.nature.com/articles/nature16039?proof=true1)

Reviewers' comments:

Reviewer's Responses to Questions

**Comments to the Author**

1. If the authors have adequately addressed your comments raised in a previous round of review and you feel that this manuscript is now acceptable for publication, you may indicate that here to bypass the “Comments to the Author” section, enter your conflict of interest statement in the “Confidential to Editor” section, and submit your "Accept" recommendation.

Reviewer #1: All comments have been addressed

Reviewer #2: (No Response)

2. Is the manuscript technically sound, and do the data support the conclusions?

Reviewer #1: Yes

Reviewer #2: Partly

3. Has the statistical analysis been performed appropriately and rigorously? 

Reviewer #1: Yes

Reviewer #2: Yes

4. Have the authors made all data underlying the findings in their manuscript fully available?

Reviewer #1: Yes

Reviewer #2: No

5. Is the manuscript presented in an intelligible fashion and written in standard English?

Reviewer #1: Yes

Reviewer #2: (No Response)

6. Review Comments to the Author

Reviewer #1: The authors have addressed most of the comments, however, for clarity of the information the minor comments below needs to be addressed before publishing.

Additional Minor concerns:

I understand the author’s focus is on timing of IRS with benthiocarb and Actellic, however, I still think there is a missing link on how the author has addressed the question 1 linked to paragraph one in the introduction section. In my opinion not only non-pyrethroids have produced contrasting impacts.

In the methodology section, the author has addressed that studies have shown that LLIN may have effect yet only referenced 1 article. I also think the statement requires a bit of modification to suit its placement in the methodology section, otherwise the author has mentioned a similar comment in the discussion section, under limitation so either way do away with it in methodology section or rephrase.

Reviewer #2: Please see the attached response to authors

I have indicated that not all data are provided but only because the baseline data are not shown and these would be useful. Thank you for providing the remaining data

I have some concerns on using the monthly (or longer) estimates of rainfall and EIR for associations with timing of spraying. This may not be a fine enough resolution to usefully determine an association and may explain why none were found.

I have some concerns on the interpretation and conclusions as a consequence. These are addressed in the attached review.

7. PLOS authors have the option to publish the peer review history of their article (what does this mean?). If published, this will include your full peer review and any attached files.

Reviewer #1: No

Reviewer #2: No

---

## [Author Response · Author response to Decision Letter 1]

25 Aug 2020

To: The Editor-in-Chief

PLOS ONE

Reference number: PONE-D-20-03773

Thank you very much for the thoughtful reviews of our manuscript and for inviting us to re-submit our revised manuscript (based on the reviewers’ comments) titled: “Estimating the Optimal Interval between Rounds of Indoor Residual Spraying of Insecticide Using Malaria Incidence Data from Cohort Studies”. Below are point by point responses to the reviewers’ comments. We confirm that the material in this manuscript has not and will not be offered elsewhere for possible publication, as long as it is under PLOS ONE consideration. We look forward to your favourable consideration of this re-submission. 

POINT BY POINT RESPONSES TO THE REVIEWERS’ COMMENTS.

Reviewer 1:

Review

Article: PONE-D-20-03773R1

Title: Estimating the Optimal Interval between Rounds of Indoor Residual Spraying of Insecticide Using Malaria Incidence Data from Cohort Studies

Summary: The work tracks clinical incidence data for a cohort of 6-month to 10-year olds living in a high transmission setting that has implemented IRS. During Phase I, the study explores the use of bendiocarb sprayed approximately bi-annually. During phase II, the study explores annual implementation with Actellic 300®CS. The limitations of the study are better explained in this revision. However, I still have some major reservations.

There remain a few queries on the methods and I have some hesitations on how the results are discussed – particularly in the abstract

Thank you for providing the data, that is useful for the review process. How was the annual EIR calculated? Please add a few sentences to explain how this was done with the data available. Similarly, it looks like rainfall was estimated monthly or over longer periods – is it possible to get finer scale information on rainfall – by week perhaps? This could impact dramatically on whether it is statistically important.

Response: 

Annual EIR defined as the number of infectious bites per person per year was calculated as a product of the daily human biting rate (HBR) and the sporozoite rate × 365 days/year. The following sentence has been added in the methodology “EIR was calculated as a product of the daily human biting rate (HBR) and the sporozoite rate × 365 days/year as previously described by our group (16)”. See lines 135-136 of the manuscript. 

Unfortunately, rainfall data was received when already aggregated by month and therefore we could not revise the results using weekly data. A discussion of this has been added as follows “For rainfall and EIR, perhaps it is because the data we used were already aggregated by month. Finer intervals like weekly data would help to improve our estimates.” See lines 272-274 of the manuscript.

The methods first focus on the statistical methods, but it may be more intuitive to present the study area, data collection and PRISM trial prior to the statistics to help the reader follow the work reported.

Response: This has been revised as suggested. See lines 96-161 of the manuscript.

Have you tried to include a variable like ‘month since net distribution’ or even a binary ‘net distribution within 1 year’ variable to try to account for the distinct impact seen from new nets?

Response: A variable for months since net distribution has been included in the analysis. However, this did not improve the model fit (p=0.249). Note that nets were distributed at enrolment and there was a universal distribution in May 2017. This result has been presented. See lines 200-201 of the manuscript.

The authors present, in the abstract, the clinical incidence from October 2014 and February 2019. But this could be driven by seasonality. It would be better to compare two times from the same month of the year, or even better still, averaging clinical incidence across a wider window of time to mitigate somewhat for natural between-year stochasticity in the metric. For how many months were the children tracked prior to application of the spray? Could you average across this entire period and compare it to the same length of time (and months) in the post application periods? Can you also provide the baseline data in the supplement?

I have the same issue with Table 1.

Table 1 is not so useful as it currently stands – it could be more informative to average or sum clinical incidence in the tracked cohorts across longer time windows (matching months) as, although seasonal rainfall did not appear to be associated with the duration of IRS effectivity in the GAM analysis, seasonal patterns in malaria incidence are likely and do tend to drive transmission by increasing the EIR, so comparing different times of the year may be misleading. Again, averaging or incidence over 3 or 6 months at baseline and comparing that to the same months in later years would be a fairer presentation. This seems particularly important because Fig 1 seems to suggest a higher mean incidence in Oct 2016 (than the baseline Oct 2014 estimate) but low in Dec 2016 – which is chosen as the comparison estimate … this is not a fair comparison so estimating the average clinical incidence across multiple months to some extent could overcome this potential stochasticity in measurements and alleviate the need to arbitrarily select a single measurement for the table.

Response: Thank you so much for this observation and for the suggestions. We have revised the presentation of our results in Table 1 by looking at the 6-months period before IRS (baseline) compared to the 6-months periods post IRS application. The periods we considered are Nov2013-Apr2014, May2014-Oct2014, Nov2014-Apr2015, May2015-Oct2015, Nov2015-Apr2016, May2016-Oct2016, Nov2016-Apr2017, May2017-Oct2017, Nov2017-Apr2018 and May2018-Oct2018. The incidence declined from 2.96 per person years during the baseline period (May-October 2014) to 1.74 during the first period following the first round of IRS application (November 2014-April 2015) and then to 0.02 during the last period following the 6th round of IRS. See lines 177-186 of the manuscript.

I have also a suggestion for the abstract as I think that the methods/results and particularly conclusions paragraphs could be more useful if altered slightly.

• Use the IRS product (presumably Ficam® and Actellic 300®CS) or the insecticide active ingredient (bendiocarb and pirimiphos methyl) not one of each [this issue applies throughout the manuscript]

Response: Thank you for this observation. Consistency has been obtained throughout the manuscript by using insecticide active ingredient names.

• Present the empirical data first, then the statistics

Response: Empirical data is now presented before statistics. See line 37 of the manuscript.

• Refer to the model as a statistical approach (as it is not a predictive model as the mechanism of transmission is not determined), so statistical approach is clearer.

Response: This has been revised as suggested. See lines 29 and 50 of the manuscript.

• My greatest hesitation is with the conclusion. Here, a statistical model is applied which is useful to capture the direction of an effect or compare the two IRS products, but it is not useful for future projections because it does not capture the mechanism of impact. So, inferring from the statistical model needs to be done with care. Drawing contrasts between the spray applications is reasonable, but perhaps soften how this is stated so as not to determine future application.

Below is a suggestion using tracked changes [comments to address/delete!]:

“Background : Indoor residual spraying (IRS) reduces vector densities and malaria transmission, however, the most effective spraying intervals for IRS have not been well established. We estimated the optimal timing interval for IRS using a statistical approach.

Methods : Six rounds of IRS were implemented in Tororo District, a historically high malaria transmission setting in Uganda, during the study period (3 rounds with Ficam (bendiocarb active ingredient): December 2014 to December 2015, and 3 rounds with Actellic 300®CS (pirimiphos methyl active ingredient): June 2016 to July 2018). Generalized additive models were used to estimate the optimal timing interval for IRS based on the predicted malaria incidence. The model was fitted to clinical incidence data from a cohort of children aged 0.5–10 years from selected households observed throughout the study period. Annual entomological inoculation rates were estimated [how] and aggregated monthly [looking at the data file it looks like these were averaged?] .

Results : Monthly incidence of malaria from October 2014 to February 2019 decreased from 3.25 to 0.0 per person-years in the children under 5 years, and 1.57 to 0.0 for 5-10 year-olds [this is not appropriate, please see comments above]. The optimal time interval for IRS differed between bendiocarb and pirimiphos methyl and by IRS round. To retain an optimum impact, Ficam® would require respraying 17 weeks after application whereas Actellic 300®CS could remain impactful for 40 weeks, although in the final year this estimate 36 weeks. However, we could not estimate from the data the optimal time after the second and third rounds of bendiocarb and after the second round of actellic. Neither the amount of rainfall nor the EIR were found to be statistically significant for determining the time period between spray rounds [although these are aggregated by month(?) so the data may lack the scale needed to identify impact].

Conclusion: In our setting, the effect of the two IRS products was distinct. Statistically, Actellic 300®CS provided a longer window of protection than Ficam®, although impact varied between different spray rounds and years which was not explained by rainfall or EIR in our statistical approach. Understanding the effectiveness of IRS and how long it lasts can help for planning campaigns, but one should consider the financial cost and insecticide resistance. Monitoring the timing of spray campaigns using clinical incidence could be repeated in future programs to help determine the average period of protectivity of these products.”

Response: Thank you so much for these suggestions. The abstract has been revised using your suggested edits. However, we preferred using active ingredient names instead of trade names. See lines 27-53 of the manuscript.

There is no discussion on the timing of the intervention relative to the transmission peak. Generally, Uganda has a higher transmission peak in May/June time, so perhaps the November spray rounds have artificially bigger impacts due to fewer cases / less transmission in these periods of the year.

Response: Thank you so much for this observation. The following sentence acknowledging transmission peaks in the months of May-June has been added “We also observe transmission peaks in the months of May and June which coincided with the second, fourth, fifth and the sixth rounds of IRS.” See lines 190-192 of the manuscript. 

A discussion on November spray rounds has been added in the discussion as a possible explanation for the increase in the incidence after IRS as follows: “This could be due to poor timing of IRS relative to the transmission peak. For example, our results indicate higher transmission peak in May/June time, so perhaps the November spray rounds may have artificially bigger impacts due to less transmission in these periods of the year”. See lines 267-270 of the manuscript.

Thank you for adding the sentence on fever and blood smear positivity. Just to clarify, did the blood smear have to be positive for a clinical case to be scored? Or were some patients classed to be malaria positive by fever only? This could alter assumptions.

Response: Yes, all malaria cases were first confirmed by a positive blood smear. A case of malaria was defined as having a fever (tympanic temperature > 38.0°C or history of fever in the previous 24 hours) and a positive thick blood smear was by light microscopy.

It may be worth including a sentence or two on what it means for the analysis to miss asymptomatic cases and what proportion of the population may be asymptomatic (a good paper on this issue e.g. Lindsay Wu et al Nature 2015: https://www.nature.com/articles/nature16039?proof=true1)

Response: Although the data on asymptomatic cases is available for this cohort, we used clinical malaria as the outcome as these are the individuals who would routinely be identified at the health facilities and are the main group that would drive policy decisions in an endemic setting. The prevalence of asymptomatic parasitemia was estimated at 58% in October 2014 (baseline) and 7% in February 2019 (end line). Details of the prevalence of parasitemia by age-group has been published elsewhere (Nankabirwa et al. 2020). The limitation section has been updated to capture this as follows. “Thirdly, our analysis did not consider asymptomatic infections which could possibly alter our estimates. For example, in the study setting the prevalence of asymptomatic infection was estimated at 58% in October 2014 (baseline) and 7% in February 2019 (end line)(16)”. See lines 278-281 of the manuscript.

Reviewer 2: 

Additional Minor concerns:

I understand the author’s focus is on timing of IRS with benthiocarb and Actellic, however, I still think there is a missing link on how the author has addressed the question 1 linked to paragraph one in the introduction section. In my opinion not only non-pyrethroids have produced contrasting impacts.

Response: Non-pyrethroids has been dropped and the sentence now read as follows: “However, some IRS products have contrasting impacts(8).” See line 64 of the manuscript.

In the methodology section, the author has addressed that studies have shown that LLIN may have effect yet only referenced 1 article. I also think the statement requires a bit of modification to suit its placement in the methodology section, otherwise the author has mentioned a similar comment in the discussion section, under limitation so either way do away with it in methodology section or rephrase.

Response: The manuscript has been revised: 1) to add new references that have assessed the impact of LLINs on malaria burden, 2) a variable looking at months since distribution of LLINs has been included in the model, although this did not improve the model fit (see lines 200-201 of the manuscript), 3). The statement on LLINs under methodology has been revised as follows “Studies have shown an additional effect of LLINs over IRS(15). The effect of LLINs on the IRS timing interval has been assessed in this analysis.” See lines 107-108 of the manuscript.

Reviewer #2: Please see the attached response to authors

I have indicated that not all data are provided but only because the baseline data are not shown and these would be useful. Thank you for providing the remaining data

Response: Baseline data is now analysed looking at 2 six-months periods before IRS (see Table 1). The remaining selected baseline data are shared.

I have some concerns on using the monthly (or longer) estimates of rainfall and EIR for associations with timing of spraying. This may not be a fine enough resolution to usefully determine an association and may explain why none were found.

Response: Thank you for this information, unfortunately the rainfall data available is already aggregated by month. We appreciate that this could be a long period to determine the associations with malaria incidence. This limitation has been acknowledged in the discussion as below: “The findings show that the timing interval was not altered by the amount of rainfall, the entomological inoculation rate (EIR), and the duration in months from when nets were distributed. For rainfall and EIR, perhaps it is because the data we used were already aggregated by month. Finer intervals like weekly data would help to improve our estimates.” See lines 271-275 of the manuscript.

---

## [Decision Letter · Decision Letter 2]

24 Sep 2020

PONE-D-20-03773R2

Estimating the Optimal Interval between Rounds of Indoor Residual Spraying of Insecticide Using Malaria Incidence Data from Cohort Studies

PLOS ONE

Dear Dr. Mugenyi,

Thank you for submitting your manuscript to PLOS ONE. After careful consideration, we feel that it has merit but does not fully meet PLOS ONE’s publication criteria as it currently stands. Therefore, we invite you to submit a revised version of the manuscript that addresses the points raised during the review process.

There are only a few minor revisions to make - you are almost there! Well done. Once you've responded to these minor comments from Reviewer 2 your manuscript will be accepted.

We look forward to receiving your revised manuscript.

Kind regards,

David A. Larsen, PhD

Academic Editor

PLOS ONE

Additional Editor Comments (if provided):

Please see some very minor revisions suggested from reviewer 2. Once those are addressed your manuscript will be accepted. Nearly there! And well done!

Reviewers' comments:

Reviewer's Responses to Questions

**Comments to the Author**

1. If the authors have adequately addressed your comments raised in a previous round of review and you feel that this manuscript is now acceptable for publication, you may indicate that here to bypass the “Comments to the Author” section, enter your conflict of interest statement in the “Confidential to Editor” section, and submit your "Accept" recommendation.

Reviewer #1: All comments have been addressed

Reviewer #2: All comments have been addressed

2. Is the manuscript technically sound, and do the data support the conclusions?

Reviewer #1: Yes

Reviewer #2: Yes

3. Has the statistical analysis been performed appropriately and rigorously? 

Reviewer #1: Yes

Reviewer #2: Yes

4. Have the authors made all data underlying the findings in their manuscript fully available?

Reviewer #1: Yes

Reviewer #2: Yes

5. Is the manuscript presented in an intelligible fashion and written in standard English?

Reviewer #1: Yes

Reviewer #2: Yes

6. Review Comments to the Author

Reviewer #1: The authors have sufficiently addressed the questions of concern, and in so doing it merits publication.

Reviewer #2: Thank you for re-submission. Please see the attached comments also copied below

The work tracks clinical incidence data for a cohort of 6-month to 10-year olds living in a high transmission setting that has implemented IRS. During Phase I, the study explores the use of bendiocarb sprayed approximately bi-annually. During phase II, the study explores annual implementation with Actellic 300®CS. The draft is improved on the previous versions and the previous comments are addressed well. I have a few minor comments to add and a couple of suggestions to pass on to the authors.

Introduction

There is a reference for the statement in lines 68-70: (https://malariajournal.biomedcentral.com/articles/10.1186/s12936-020-3102-6)

This sentence, starting line 70, is not quite correct as thatch is mentioned twice – please correct:

“A study in Zimbabwe showed that bendiocarb remains active for up to 8 weeks with 96% mortality for mosquitoes on thatch, and 20 weeks with 74% mortality for mosquitoes on mud compared to 100% for mosquitoes on thatch (10).”

Similarly the following sentence needs to be adjusted slightly for clarity:

“While in Madagascar, bendiocarb was shown to have an efficacy of up tomaintain 80% mosquito mortality for up to 5 months post spraying (11).”

Throughout this second paragraph of the introduction, efficacy is used without defining exactly what is meant. Perhaps state that a mortality inducing effect where more than 80% of susceptible or local mosquitoes are killed in a bioassay test on the sprayed surfaces is considered effective by WHO guidelines [ref]. (In my opinion, if sprays are killing mosquitoes at a less potent level they may well still have robust impact on incidence – but this is not the WHO definition at the moment!) Otherwise you could stick to quantifying this effect by stating that the mortality inducing effect on local mosquitoes remained above 80% for xx weeks as noted by the referenced studies.

One interesting part of your study is that your tested outcome is a clinical one. Testing IRS on mosquito mortality alone forces us to make an assumption about how reduced mosquitoes may lead to reduced clinical incidence. Your approach allows you to test the durability of the spray campaigns on the clinical outcome. I think this is worth highlighting, either as a sentence in the introduction or discussion.

It is also worth noting that malaria transmission in Uganda is mostly perennial, so even though there are slight seasonal peaks, transmission continues all year making the durability of the IRS product an important consideration. This helps understand the analysis too because the 6-month periods can be more consistently compared than would be the case in an area that had more seasonal transmission.

Discussion

Line 259 should be ‘concurs’ not ‘concur’

On line 264 perhaps say ‘may leave communities unprotected’ rather than may render IRS ineffective

Lines 269- Perhaps say ‘due to the natural decline in transmission observed during these months of the year’.

On line 274 I would simply add – “but were not available.”

Perhaps add as a limitation that you assume no change in the level of insecticide resistance to either product throughout the time-series (if you have susceptibility bioassays for the region perhaps you could corroborate this assumption showing or referencing those).

I think there is a policy suggestion from your analysis that you could reasonably add to the discussion –

This work highlights well the variability in the length of time that the active ingredient deployed during a spray campaign may have, even for the same product within a single setting. Real-time surveillance of trends in clinical incidence could be used to help advise strategy teams on when to re-deploy the IRS intervention, ensuring rotation of products as guided by insecticide resistance management policies [you can ref WHO].

The only other comment I have is that spray campaigns sometimes take months to deliver to a community, so perhaps it is worth noting the length of time, in these communities, that it took to complete delivery of each spray campaign and the ultimate percentage of households that were then covered in each spray round. Does this make a difference to the predictions?! Perhaps beyond the scope of the current paper as it is nearly ready to accept, but something to consider in future perhaps.

7. PLOS authors have the option to publish the peer review history of their article (what does this mean?). If published, this will include your full peer review and any attached files.

Reviewer #1: **Yes: **Mercy Opiyo

Reviewer #2: No

---

## [Author Response · Author response to Decision Letter 2]

6 Oct 2020

POINT BY POINT RESPONSES TO THE REVIEWER’S COMMENTS.

Review

Article: PONE-D-20-03773R2

Title: Estimating the Optimal Interval between Rounds of Indoor Residual Spraying of Insecticide Using Malaria Incidence Data from Cohort Studies

Summary: The work tracks clinical incidence data for a cohort of 6-month to 10-year olds living in a high transmission setting that has implemented IRS. During Phase I, the study explores the use of bendiocarb sprayed approximately bi-annually. During phase II, the study explores annual implementation with Actellic 300®CS. The draft is improved on the previous versions and the previous comments are addressed well. I have a few minor comments to add and a couple of suggestions to pass on to the authors. 

Introduction

There is a reference for the statement in lines 68-70: (https://malariajournal.biomedcentral.com/articles/10.1186/s12936-020-3102-6)

Response: Thank you so much for the suggested reference. This has been added. See lines 68-70

This sentence, starting line 70, is not quite correct as thatch is mentioned twice – please correct: 

“A study in Zimbabwe showed that bendiocarb remains active for up to 8 weeks with 96% mortality for mosquitoes on thatch, and 20 weeks with 74% mortality for mosquitoes on mud compared to 100% for mosquitoes on thatch (10).”

Response: Thank you for this observation. The sentence has been revised to read as follows: “A study in Zimbabwe showed that bendiocarb remains active for up to 8 weeks with 96% mortality for mosquitoes on thatch(11). The same study showed that bendiocarb remains active for up to 20 weeks with 74% mortality for mosquitoes on mud compared to 100% for mosquitoes on thatch(11).” See lines 71-74

Similarly the following sentence needs to be adjusted slightly for clarity: 

“While in Madagascar, bendiocarb was shown to maintain 80% mosquito mortality for up to 5 months post spraying (11).”

Response: The sentence has been revised to read as follows: “In Madagascar, bendiocarb was shown to have a mortality inducing effect on local mosquitoes of up to 80% for up to 5 months post spraying (12).” See lines 74-76.

Throughout this second paragraph of the introduction, efficacy is used without defining exactly what is meant. Perhaps state that a mortality inducing effect where more than 80% of susceptible or local mosquitoes are killed in a bioassay test on the sprayed surfaces is considered effective by WHO guidelines [ref]. (In my opinion, if sprays are killing mosquitoes at a less potent level they may well still have robust impact on incidence – but this is not the WHO definition at the moment!) Otherwise you could stick to quantifying this effect by stating that the mortality inducing effect on local mosquitoes remained above 80% for xx weeks as noted by the referenced studies. 

Response: Thank you for this observation and suggestion. We are happy to consider your suggested revision by replacing efficacy with “mortality inducing effect on local mosquitoes”. The affected sentences have been revised to read as follows: “In Madagascar, bendiocarb was shown to have a mortality inducing effect on local mosquitoes of up to 80% for up to 5 months post spraying (12). Elsewhere in Zanzibar, a study to investigate the residual effect of pirimiphos methyl active ingredient sprayed on common surfaces of human dwellings showed that its mortality inducing effect on local mosquitoes was maintained on all sprayed surfaces up to 8 months post-IRS (13).” Lines 74-78.

One interesting part of your study is that your tested outcome is a clinical one. Testing IRS on mosquito mortality alone forces us to make an assumption about how reduced mosquitoes may lead to reduced clinical incidence. Your approach allows you to test the durability of the spray campaigns on the clinical outcome. I think this is worth highlighting, either as a sentence in the introduction or discussion.

Response: Thank you for this important suggestion. The following sentence has been added in the last paragraph of the introduction “The use of incidence data allows us to test the durability of the spray campaigns on the clinical outcome.” Lines 90-91.

It is also worth noting that malaria transmission in Uganda is mostly perennial, so even though there are slight seasonal peaks, transmission continues all year making the durability of the IRS product an important consideration. This helps understand the analysis too because the 6-month periods can be more consistently compared than would be the case in an area that had more seasonal transmission. 

Response: This is a true observation. A sentence noting this has been added under the study setting as follows: “Malaria transmission in Uganda is mostly perennial, so even though there are slight seasonal peaks, transmission continues all year making the durability of the IRS product an important consideration.” Lines 103-105.

Discussion

Line 259 should be ‘concurs’ not ‘concur’

Response: This is done. The sentence now reads as follows: “However, for pirimiphos methyl, the results suggest that about one round per year is feasible and this concurs with the WHO recommendations(14).” See lines 263-265. 

On line 264 perhaps say ‘may leave communities unprotected’ rather than may render IRS ineffective

Response: This is done. The sentence now reads as follows: “This implies that waiting for 7 months in case of bendiocarb before another round of IRS may leave communities unprotected.” See lines 268-269. 

Lines 269- Perhaps say ‘due to the natural decline in transmission observed during these months of the year. 

Response: This is done. The sentence now reads as follows: “For example, our results indicate higher transmission peak in May/June time, so perhaps the November spray rounds may have artificially bigger impacts due to the natural decline in transmission observed during these months of the year.” See lines 273-275. 

On line 274 I would simply add – “but were not available.”

Response: This is done. The sentence now reads as follows: “Finer intervals like weekly data would help to improve our estimates but were not available.” See lines 278-279.

Perhaps add as a limitation that you assume no change in the level of insecticide resistance to either product throughout the time-series (if you have susceptibility bioassays for the region perhaps you could corroborate this assumption showing or referencing those).

Response: This is done by adding a fourth limitation as follows: “Fourth, we assume no change in the level of insecticide resistance to either product throughout the time-series. We did not have susceptibility bioassays for the study region to corroborate this assumption.” Lines 291-293.

I think there is a policy suggestion from your analysis that you could reasonably add to the discussion –

This work highlights well the variability in the length of time that the active ingredient deployed during a spray campaign may have, even for the same product within a single setting. Real-time surveillance of trends in clinical incidence could be used to help advise strategy teams on when to re-deploy the IRS intervention, ensuring rotation of products as guided by insecticide resistance management policies [you can ref WHO].

Response: Thank you so much for this suggested discussion. A paragraph has been added in the discussion as follows: “This work highlights well the variability in the length of time that the active ingredient deployed during a spray campaign may have, even for the same product within a single setting. Real-time surveillance of trends in clinical incidence could be used to help advise strategy teams on when to re-deploy the IRS intervention, ensuring rotation of products as guided by insecticide resistance management policies(14).” Lines 280-284.

The only other comment I have is that spray campaigns sometimes take months to deliver to a community, so perhaps it is worth noting the length of time, in these communities, that it took to complete delivery of each spray campaign and the ultimate percentage of households that were then covered in each spray round. Does this make a difference to the predictions?! Perhaps beyond the scope of the current paper as it is nearly ready to accept, but something to consider in future perhaps.

Response: A sentence showing length of time in months it took to complete each round of IRS is added under the study setting section as follows: “The length of time in months it took to complete delivery of each spray campaign were 1.9 for round 1, 1.3 for round 2, 0.8 for round 3, 1.0 for round 4, 2.5 for round 5, and 1.3 for round 6.” See lines 109-111. Unfortunately, we could not estimate the ultimate percentage of households that were covered because data on household listing were not available. Yes, it’s likely that the duration of completing each round of IRS could affect the prediction. This analysis will be considered in the future.

---

## [Editor Report · Decision Letter 3]

8 Oct 2020

Estimating the Optimal Interval between Rounds of Indoor Residual Spraying of Insecticide Using Malaria Incidence Data from Cohort Studies

PONE-D-20-03773R3

Dear Dr. Mugenyi,

We’re pleased to inform you that your manuscript has been judged scientifically suitable for publication and will be formally accepted for publication once it meets all outstanding technical requirements.

Kind regards,

David A. Larsen, PhD

Academic Editor

PLOS ONE

Additional Editor Comments (optional):

Well done!
---

## [Editor Report · Acceptance letter]

14 Oct 2020

PONE-D-20-03773R3 

Estimating the Optimal Interval between Rounds of Indoor Residual Spraying of Insecticide Using Malaria Incidence Data from Cohort Studies 

Dear Dr. Mugenyi:

I'm pleased to inform you that your manuscript has been deemed suitable for publication in PLOS ONE. Congratulations! Your manuscript is now with our production department. 

Kind regards, 

on behalf of

Dr. David A. Larsen 

Academic Editor

PLOS ONE